# Investigating mental representations of psychoactive substance use and other potentially addictive behaviors using a data driven network-based clustering method

**Domonkos File** [1]*, **Bálint File**[2], **Beáta Bőthe**[3], **Mark D. Griffiths**[4], **Zsolt Demetrovics**[1,5]

**1** Institute of Psychology, ELTE Eötvös Loránd University, Budapest, Hungary, **2** Wigner Research Centre for Physics, Budapest, Hungary, **3** Département de Psychologie, Université de Montréal, Montreal, Canada, **4** Psychology Department, International Gaming Research Unit, Nottingham Trent University, Nottingham, United Kingdom, **5** Centre of Excellence in Responsible Gaming, University of Gibraltar, Gibraltar, Gibraltar

* file.domonkos@ppk.elte.hu

**Data Availability Statement:** Data is available at https://zenodo.org/record/7956993.

## Abstract

### Background and aims

The aim of the present study was to examine the mental representations of the use of different substances and other potentially addictive behaviors in order to explore meaningful similarities and differences that may contribute to a better understanding of behavioral addictions' representations and diagnostic criteria.

### Methods

The authors mapped the mental and emotional representations of 661 participants (70.5% women; $M_{age}$ = 35.2 years, SD = 11.7) to the concept "your most disturbing excessive activity" using free-word associations combined with a network-based clustering method.

### Results

The network analyses identified four distinct mental representations, three implicating dominantly negative (Guilt/Shame/Relief, Addiction/Health, and Procrastination/Boredom) and one dominantly positive emotion (Stress/Relaxation). The distribution of Addiction/Health and Procrastination/Boredom representations were different across substance use and problem behaviors, indicating meaningful differences in the underlying cognitive evaluation processes. The Addiction/Health representation was more frequent for substances, while for other addictive behaviors, the Procrastination/Boredom representation was more frequent, and its frequency increased with the self-reported intensity of the behavior. Guilt/Shame/Relief was equally common for both substances and behaviors, but importantly, for substances its' likelihood increased with the intensity of use.

### Conclusion

The common part of representations for substance use and other potentially addictive behaviors supports the scientific viewpoint, that real addictions can exist even in the

**Funding:** The study was supported by the National Research, Development and Innovation Office (K126835, K131635, PD138976). BB was supported by a postdoctoral fellowship from the SCOUP Team – Sexuality and Couples – Fonds de recherche du Québec, Société et Culture. The funders had no role in study design, data collection and analysis, decision to publish, or preparation of the manuscript.

**Competing interests:** ELTE Eötvös Loránd University receives funding from the Szerencsejáték Ltd. to maintain a telephone helpline service for problematic gambling. ZD has also been involved in research on responsible gambling funded by Szerencsejáték Ltd. and the Gambling Supervision Board and provided educational materials for the Szerencsejáték Ltd's responsible gambling program. The University of Gibraltar receives funding from the Gibraltar Gambling Care Foundation. MDG's university has received research funding from Norsk Tipping (the gambling operator owned by the Norwegian Government). MDG has also received funding for a number of research projects in the area of gambling education for young people, social responsibility in gambling and gambling treatment from Gamble Aware (formerly the Responsible Gambling Trust), a charitable body which funds its research program based on donations from the gambling industry. MDG regularly undertakes consultancy for various gaming companies in the area of social responsibility in gambling. However, these funding sources are not related to the present study and the funding institution had no role in the study design or the collection, analysis, and interpretation of the data, writing the manuscript, or the decision to submit the paper for publication. This does not alter our adherence to PLOS ONE policies on sharing data and materials.

absence of psychoactive drugs. Based on the results, a novel proposition is posited, that a more appropriate indicator of tolerance for problem behaviors might be the perceived amount of time wasted on the activity rather than the actual time spent.

## Introduction

Growing evidence suggests that behavioral addictions (e.g., addictions to gambling, gaming, sex, etc.) share significant similarities with substance use addictions in many domains, such as the etiology, tolerance, comorbidity, and response to treatment [1–5] Besides the similarities, consensus about the applicability of some factors has not been reached such as physical signs of addiction [6] and tolerance [7,8].

A major critique against behavioral addiction research is that the diagnostic components are inherited from substance addiction practice, often leading to the negligence of the heterogeneity of these disorders [9]. For example, tolerance, one of the key features of addiction, is a highly debated criterion [10,11]. As Billeux et al. (2015b) illustrate, no-one would argue if someone started to play the guitar and spends more and more time with it, developing tolerance towards the behavior and/or "music addiction". Similarly, mood modification has low specificity in case of internet gaming disorder [12] and problematic pornography use [13] (i.e., a significant proportion of non-addicted users also engage in the activity to modify their mood). As Griffiths [10] points out, evaluating the balance of positive and negative consequences is necessary to distinguish between excessive enthusiasms and addictions. If positive consequences outweigh negatives, the behavior in question cannot be defined as an addiction. To define such a balance, a multivariate function is required, for which the exploration of possible dimensions with high explanatory power can increase the accuracy of diagnosis. Also, considering the lack of consensus concerning the common diagnostic components, and the often not straightforward interpretability in case of behavioral addictions [9,14,15], introducing novel empirical approaches to further explore differences and similarities might be beneficial for either scientific purposes or clinical use in the long-term.

A potentially useful approach to acquire the adequate frames is to map the structure and content of mental representations (MRs) towards their own addiction from individuals with behavioral or substance addiction. A widely used tool to examine MRs is the free word association (FWA) task, through assessing the first ideas elicited by a cue (stable implicit attitudes by verbal associations). FWAs are considered as a technique for overcoming the limitations of predetermined frames by exploring representations beyond the researcher's preconceptions. Similar to open-ended questions, the FWA task allows participants to describe their beliefs and experiences in their own words [16]. Although classical open-ended questions provide more detailed information than FWAs, the higher complexity of texts require more advanced natural language techniques, which are still not well-established measures [17,18]. Some researchers even argue that these answers assess the ability of the participant to articulate the sentences and not their opinions [19]. It has also been suggested that FWAs related to engagement in cue-relevant behaviors [20] because past behaviors generate associations in memory that bias future behavior. Given the potential of the FWA task to provide unrestricted access to MRs, their exploration may help to understand the mechanisms underlying behavioral and substance addictions.

Addiction research using FWA techniques began with the work of [21] Szalay et al. (1992), who focused on the representational differences of individuals with and without substance use disorder, regarding both substance-specific topics ("drug" [22]) and general topics (e.g., "me",

"family", "father" [21]). They reported meaningful differences in both perceptual and motivational dispositions between pretreatment and rehabilitated clients, with possible consequences on treatment evaluation [21,22]. Stacy et al. [23] presented a series of alcohol-related and neutral short phrases to college students who were asked to generate the first behavior that came to mind. Strong associations were found between the generation of alcohol responses and alcohol consumption, indicating the importance of memory processes in alcohol use in a non-clinical sample [23]. Reich, Goldman & Noll [24] investigated alcohol-related representations on a large college sample. They found that heaviest alcohol drinkers had more positive and arousing responses than did lighter alcohol drinkers, who had more negative and sedating responses. The results were also interpreted as evidence of the influence of memory processes on the behavioral response to alcohol consumption [24]. In another study, cigarette smoking-related positive and negative information was assessed with an FWA test across smokers and non-smokers [25]. They have found, that although smokers generated more positive smoking-related associations than non-smokers, in total, both groups produced more negative associations. Cigarette smokers in the early time interval generated proportionally more positive associations, suggesting that positive smoking-related memories were easier to activate, even though they also have more negative associations available [25].

Although FWA techniques are capable of identifying meaningful MRs explaining addiction, research using this technique has remained scarce in this field. In addition, these techniques have used human raters to identify relevant FWA groups. If FWAs are considered as a technique for overcoming the limitations of predetermined questions by exploring representations beyond the researcher's preconceptions, then the clustering of FWAs by the researcher may also hinder this goal. To overcome this limitation, there are a few clustering solutions tried by previous studies, such as factor analytic methods [26] and hierarchical agglomerative clustering [27]. The current study implemented a modular analysis of the co-occurrence network of free word associations (FWAs) [28]. We applied this approach, as our previous methodological research has demonstrated the capability of modular analysis in establishing a reproducible result [28] and connecting mental representations (MRs) to cue-specific behaviours and attitudes of participants [28,29].

With an exploratory intent, the present study investigated possible differences between the (i) MRs of potentially addictive substances and behaviors, (ii) emotions linked to the MRs of potentially addictive substances and behaviors, and (iii) structure of MRs as a function of subjective severity of the substance use and other potentially addictive behaviors.

## Materials and methods

### Participants

The samples were collected via online questionnaires, and survey completion took approximately 20 minutes. The sample used in this study may not be representative of the Hungarian population, as it was obtained through a popular Hungarian news portal and public, topic-irrelevant Facebook pages. The study was advertised as a research project concerning the psychological factors of excessive activities, and data were collected from June to September in 2020. Informed consent was obtained from participants before data collection, and participants were ensured of their anonymity. The present study was conducted adhering to the Declaration of Helsinki and was approved by the institutional ethical review board of the Eötvös Loránd University (2020/258). No personal information that allowed personal identification was asked, and a secure online platform (Qualtrics Research Suite; Qualtrics, Provo, UT) was used for data collection. Required sample size was determined based on a previous study using the same methodology [30].

**Table 1. Overview of the demographic characteristics of the sample.**

|  | % | Mean | SD | Range |
|---|---|---|---|---|
| **Age** |  | 35.2 | 11.7 | 18–74 |
| Gender (female) | 70.5 |  |  |  |
| **Completed education** |  |  |  |  |
| Primary education | 2.3 |  |  |  |
| Vocational degree | 5.9 |  |  |  |
| High-school degree | 15.3 |  |  |  |
| College or university degree | 76.4 |  |  |  |
| **Relationship Status** |  |  |  |  |
| Single | 27.9% |  |  |  |
| In romantic relationship | 69.3 |  |  |  |
| Other | 2.3 |  |  |  |

The inclusion criteria were (i) providing informed consent and (ii) being aged 18 years or older and (iii). Overall, 661 participants (466 women, 70.5%) aged between 18 and 74 years ($M_{age}$ = 35.2 years, SD = 11.73) completed the survey. Of these, 2.3% had maximum primary education, 5.9% reported having a vocational degree, a further 15.3% had high-school degree, and 76.4% college or university degree. Regarding relationship status, 185 were single (27.9%), 458 were in any kind of romantic relationship (i.e., being in a romantic relationship or married) (69.3%), 17 chose the "other" option (2.6%), and one participant did not provide a response to this question Table 1 presents the demographic characteristics of the sample.

## Procedure and measures

Participants were asked to write five words or expressions that comes into mind regarding the sentence "Imagine you are doing the excessive activity that bothers you the most". We have deliberately chosen to use "excessive activity" instead of using the word 'addiction' as a call word, as (1) abstract, often used expressions easily trigger common knowledge rather than subjectively shaped MRs, (2) the word 'addiction' is highly stigmatized [31] and might leads to resistance, and (3) different interpretations of the word would make it difficult to interpret the results. After providing all five associations, participants got back their associations one-by-one and were asked to select two emotional labels from the list of 20 Positive and Negative Affect Schedule (PANAS) labels [32] to each of the associations. Using emotions to enhance the analysis of free-word associations has proven to be of benefit in recent studies [28].

Participants were then asked to indicate the substance/behavior they thought of from a list of four psychoactive substances and ten potentially addictive behaviors (see Table 2). If the substance/behavior in question was not on the list, an "other" option could be chosen. After this, a subjective evaluation of the problem severity was indicated using one item (*"I did it too much in the past 12 months"*) taken from the Screener for Substance and Behavioral Addictions (SSBA, [33]) comprising four response options: 'Totally disagree', 'Partly disagree', 'Partly agree', 'Totally agree'.

## Analyses

Free-word associations were (i) first spellchecked (transformed to lower case, removed accents, and manually corrected), (ii) lemmatized and (iii) translated to English. Associations were merged if their English translation was identical, or was a close synonym. While assessing the substance/behavior the participants thought of, 76 (out of 661) participants indicated

**Table 2. The distribution and demographic characteristics of the investigated substance uses and behaviors.**

|  | N of responses | Severity of problematic activity | Gender of respondents (% of females) | Age of respondents |
|---|---|---|---|---|
| Alcohol use | 66 | 3.5 (0.75) | 69.7% | 34.16 (8.62) |
| Tobacco use | 87 | 3.72 (0.58) | 67.8% | 37.24 (12.45) |
| Cannabis use | 22 | 3.68 (0.57) | 45.5% | 31.22 (7.82) |
| Other substance use | 11 | 3 (1.34) | 63.6% | 28.27 (4.94) |
| Social media use | 85 | 3.6 (0.71) | 87.0% | 30.33 (10.23) |
| Internet use | 65 | 3.57 (0.66) | 76.9% | 33.4 (11.27) |
| TV series watching | 25 | 3.56 (0.65) | 84.0% | 33.16 (12.46) |
| Eating | 73 | 3.60 (0.55) | 93.15% | 36.33(10.81) |
| Work | 39 | 3.79 (0.57) | 61.5% | 39.76 (12.34) |
| Pornography use | 33 | 3.48(0.83) | 12.1% | 36.53 (12.47) |
| Shopping | 19 | 3.26(0.93) | 89.5% | 38.42 (11.64) |
| Gaming | 38 | 3.39 (0.75) | 44.7% | 37.61 (11.48) |
| Sex | 17 | 3.06 (1.08) | 47.1% | 42.58 (11.62) |
| Gambling | 5 | 2.8 (1.09) | 20.0% | 45.2 (15.99) |
| Other behaviors | 76 | - | 78.9% | 36.71 (13.85) |

'other' from the list of choices (see Table 2). As it is not known whether their associations concerned a substance or a behavior, these participants were excluded from further analysis. FWAs mentioned by less than 10 participants were excluded from the analysis as such terms can refer to unstable or idiosyncratic parts of the representations [34,35] and a breakpoint at a frequency of 10 was observable (see figure at Appendix), indicating that excluding associations mentioned less than 10 times still retained 65% of the total FWAs in the analysis. Participants whose associations' frequency for each category was lower than 10 were unable to be classified (n = 40) and were subsequently excluded from subsequent analyses. This resulted in a final study cohort consisting of 546 participants. PANAS labels were considered relevant in the context of the study that were provided by at least 50% of participants overall. The decision to set this threshold was arbitrary, serving the purpose of maintaining stability in emotional labels and directing the study towards emotional states that, due to their frequent occurrence, were more likely to have a general consensus among the respondents. S2 Appendix contains the results pertaining to the labels that were not considered due to the implementation of this approach.

The present study applied multiple FWAs (five FWAs/individuals) with a data-driven method (i.e., associations were connected base on their statistical co-occurrences [CoOp] method; see [28]]), in which FWA groups are derived from patterns of individual representations. More specifically, networks were constructed from the statistical co-occurrence of the FWAs and distinct MRs were identified as densely connected FWAs. It has been demonstrated in recent studies that MRs identified by FWAs from the aforementioned network procedure can create a stable structure and can be linked to the cue-specific behavior and attitudes of the participants [28,29].

In this analysis, major MRs of addiction were extracted from the network analysis of FWAs. In this network, each unique association was defined as a node and co-occurrence relationships between the associations as edges. Statistical co-occurrence between every pair of associations was defined as the loglikelihood ratio between the maximum likelihood of the observed co-occurrence and the likelihood assuming statistical independence. Associations were *attractive* (i.e., attained a positively weighted edge) if they were mentioned more frequently together than by chance or *repulsive* (i.e., attained a negatively weighted edge) if they were mentioned less frequently together than by chance. Higher deviation from chance

indicated higher negative or positive weight. Densely connected subnetworks were identified as modules. These modules reflecting different MRs regarding *"most disturbing activity"*. Modules were defined by consensus clustering from the maximal modularity partitions of the network identified by the Louvain algorithm [36,37]. Due to the negative edges in the network, an extended formula of modularity was applied:

$$Q = \frac{1}{v^+ + v^-} \sum_{ij} [(w_{ij}^+ - e_{ij}^+) - (w_{ij}^- - e_{ij}^-)] \delta_{M_i M_j}$$

(i) Q denotes the modularity value of a given partition of a network, (ii) $v^+/v^-$ denotes the total positive/negative weights of the network, (iii) $w_{ij}^+/w_{ij}^-$ denotes the positive/negative weights between node *I* and *j*, (iv) $e_{ij}^+/e_{ij}^-$ denotes the chance-expected positive/negative connections between node *I*, and (v) *j*, $\partial_{M_i M_j}$ is an indicator function that is set to 1 if node i and j belong to the same module.

The emotional features of the association modules were described by the emotional labels of the associations. The relative percentage distribution of the emotions in each module was calculated. To characterize the emerging modules with substances or behaviors, each participant was linked to the module from which most of their associations came from.

Chi-square tests of independence were performed to examine the relationship between misuse type (substance or behavioral) and MRs. Also, chi-square tests of independence were performed to examine the relationship between problem intensity (low, high) and MRs for substance uses and behaviors separately. Problem intensity was defined on the basis of the SSBA item (*"I did it too much in the past 12 months"*). Participants who indicated 'Totally disagree', or 'Partly disagree' or 'Partly agree' were classified into the low-intensity group, whereas participants who indicated 'Totally agree' were classified into the high-intensity group. Kruskal-Wallis H tests were performed to explore possible differences (i) between emotions of psychoactive substances and potentially addictive behaviors, and (ii) between emotions related to the four emerging representational modules.

The network construction was performed with MATLAB version R2019b [38] (The MathWorks Inc, Natick, MA). The applied network measures are available at https://sites.google.com/site/bctnet/ [39]. Visualization of the network was done by Gephi 0.9.2 [40].

## Results

The total number of unique associations was 1374. After the spellcheck, lemmatization, and merging, the number of associations decreased to 254, of which 77 were non-idiosyncratic and represented the basis of the network analysis. It is important to stress that the 77 unique association covered 65% of the total number of produced FWA answers (for further details see S1 Fig in S2 Appendix).

For the item *"I did it too much in the past 12 months"*, 3% indicated 'Totally disagree', 3% 'Partly disagree', 25.8% 'Partly agree', and 68.2% 'Totally agree'.

The network analyses identified four distinct MRs which were subsequently labeled as Guilt/ Shame/Relief, Addiction/Health, Stress/Relaxation, and Procrastination/Boredom(see Fig 1).

Six dominant emotions were covered by the 51% of the selected PANAS labels: worries, shame, anxiety, contempt, joy, and calmness. The distribution of these emotions was different across MRs ('worries' [H = 14.6, df = 3, *p* = 0.002]; 'shame' [H = 23.8, df = 3, *p*<0.001]; 'anxiety' [H = 13.7, df = 3, *p* = 0.003], 'contempt' [H = 19.9, df = 3, *p*<0.001], 'joy' [H = 65.1, df = 3, *p*<0.001], 'calmness' [H = 76, df = 3, *p*<0.001]). The outcomes of pairwise comparisons using Dunn's tests are reported in Table 3 and Fig 2.

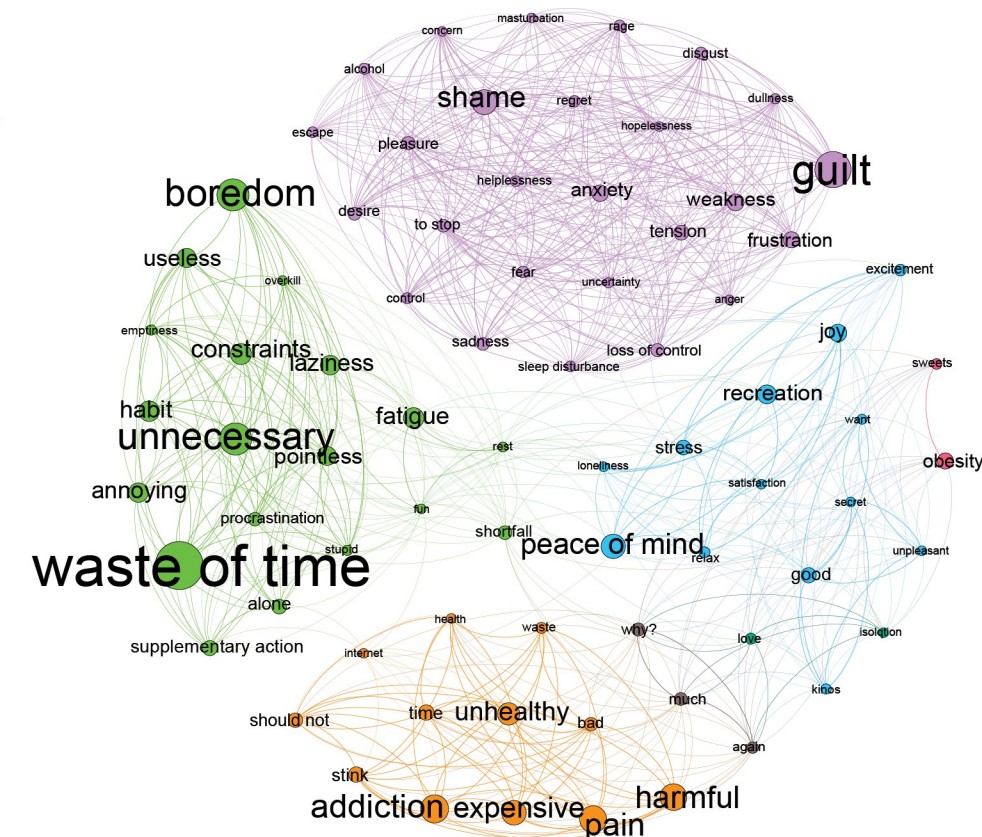

**Fig 1. Mental representations of the CoOp networks.** Each MR is visualized with different colors (purple:Guilt/ Shame/Relief; blue: Stress/Relaxation; orange: Addiction/Health, green: Procrastination/Boredom). The sizes of a node and its label are proportional to the frequency of the given association. An edge means that two associations fall into a common MR in the consensus-partitioning procedure at least 40%.

The relationship between problem type and the related MRs was significant ($\chi^2$[df = 3, N = 546] = 103.3, $p < .0001$). Associations from the Addiction/Health representation were more likely in case of psychoactive substances, while associations from the Procrastination/Bore-domrepresentation were more likely for potentially addictive behaviors (Fig 3A). The relationship between intensity and representations were significant for both psychoactive substances ($\chi^2$[df = 3, N = 180] = 85.0, $p < .0001$ and potentially addictive behaviors ($\chi^2$[df = 3, N = 366] = 36.0, $p<0.0001$ (Fig 3B and 3C). In case of high intensity substance use, associations were more frequent in the Guilt/Shame/Relief representation, while in case of low intensity

**Table 3. Pairwise comparison of emotions across mental representations.**

|  | Worries | Shame | Anxiety | Contempt | Joy | Calmness |
|---|---|---|---|---|---|---|
| Guilt/Shame/Relief-Stress/Relaxation | 0.02 | <0.001 | 0.003 | 0.003 | <0.001 | <0.001 |
| Addiction/Health-Stress/Relaxation | <0.001 | - | 0.002 | 0.02 | <0.001 | <0.001 |
| Procrastination/Boredom—Stress/Relaxation | 0.02 | 0.04 | 0.004 | <0.001 | <0.001 | <0.001 |
| Addiction/Healt-Procrastination/Boredom | - | - | - | 0.04 | - | - |
| Guilt/Shame/Relief-Procrastination/Boredom | - | 0.01 | - | - | - | - |

Numbers indicate p-values.

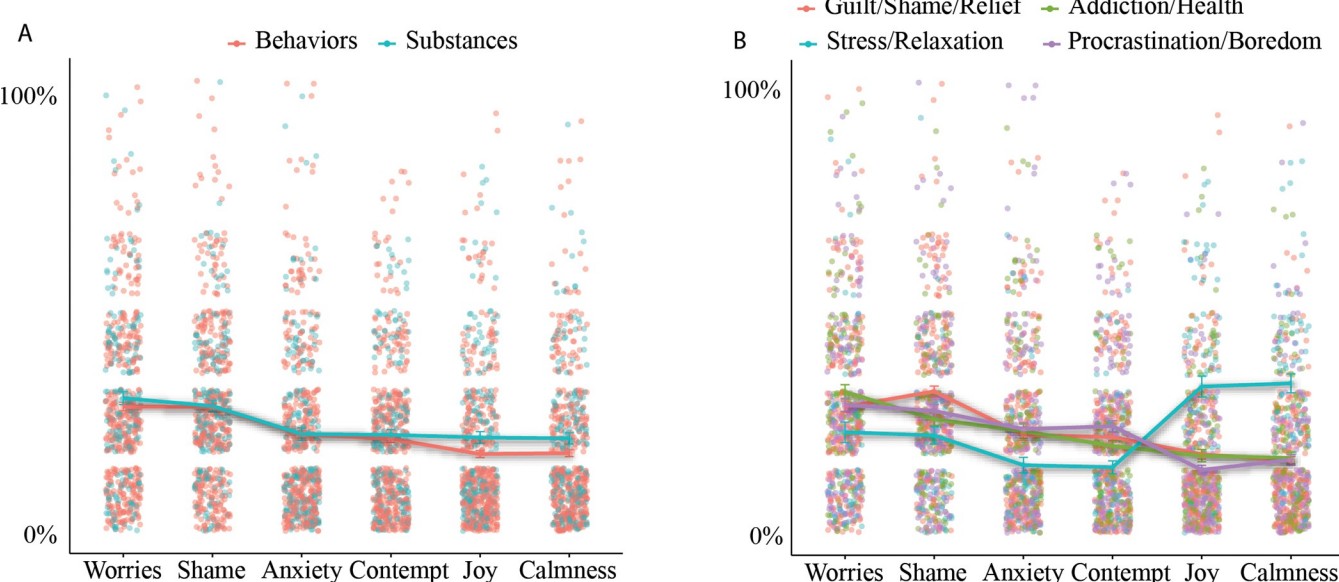

**Fig 2.** Graphical representation of the differences in emotions between (A) psychoactive substance uses and potentially addictive behaviors and (B) between MRs.

substance use associations, the Stress/Relaxation and Procrastination/Boredom representations were more frequent. In case of high intensity potentially addictive behaviors, associations were more frequent in the Procrastination/Boredom representation, while in case of low intensity problem behaviors associations, the Relaxation and Addiction/Health representations were more frequent. There was no difference regarding emotions and positive and negative affects between low-intensity and high-intensity psychoactive substance uses and potentially addictive behaviors.

Comparing emotions between psychoactive substance uses and potentially addictive behaviors across the four modules using the Kruskal-Wallis chi-squared test only identified significant differences between the positive emotions of 'joy' (H = 7.3, df = 3, $p$ = 0.006) and 'calmness' (H = 6.6, df = 3, $p<0.01$)–see Fig 2A. That is, 'joy' and 'calmness' were indicated more often for the associations of psychoactive substance uses.

## Discussion

Previously, no explorative data-driven studies have ever described the differences between the mental representations (MRs) of psychoactive substance use behaviors and other potentially addictive behaviors, despite preliminary evidence suggesting FWA's applicability and usefulness in understanding different potentially addictive behaviors (e.g., [21–23]). Therefore, the aim of the present study was to fill this gap by mapping mental and emotional representations of the concept of individuals' *"most disturbing excessive activity"* using free-word associations combined with a data-driven clustering method [28] and linking these representations to specific measures of substance uses and potentially addictive behaviors. The analysis showed that the perception of *"disturbing excessive activity"* formed four distinct MRs, three with dominantly negative emotions (Guilt/Shame/Relief, Addiction/Health and Procrastination/Boredom) and one with dominantly positive emotions (Stress/Relaxation) (see Fig 2B).

*Guilt/Shame/Relief* (35%): The most frequent MR included associations of diffuse concepts, such as "guilt", "shame", "weakness", "anxiety" and "frustration" which suggests generalized

**A**

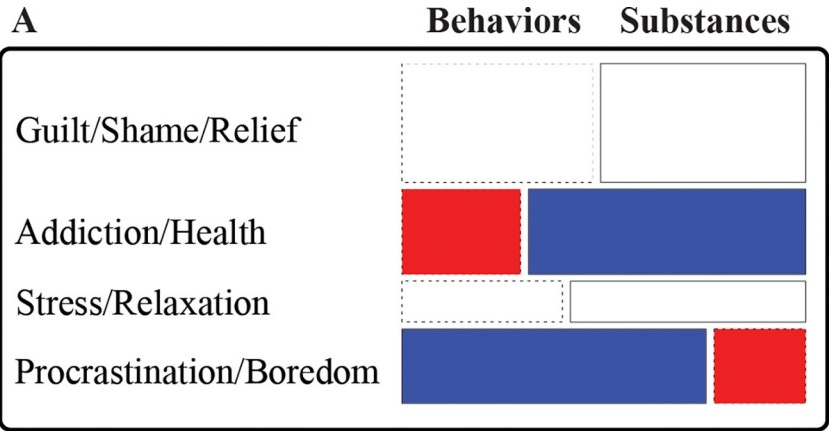

**B**

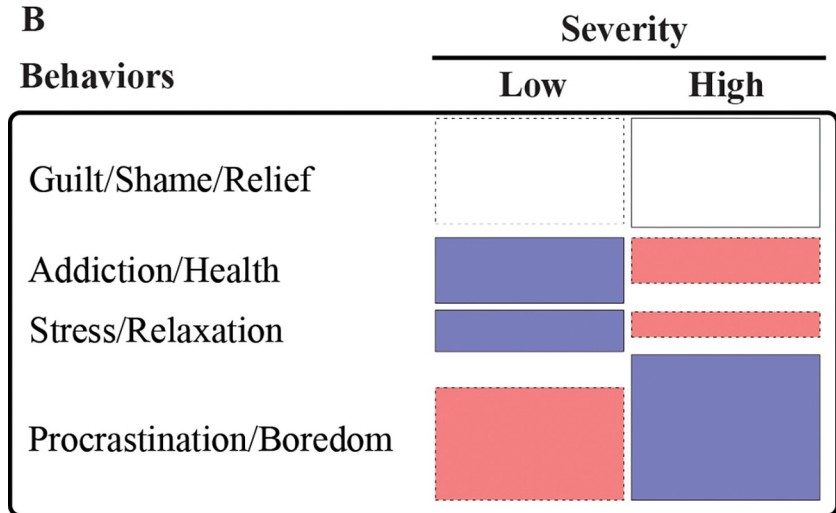

**C**

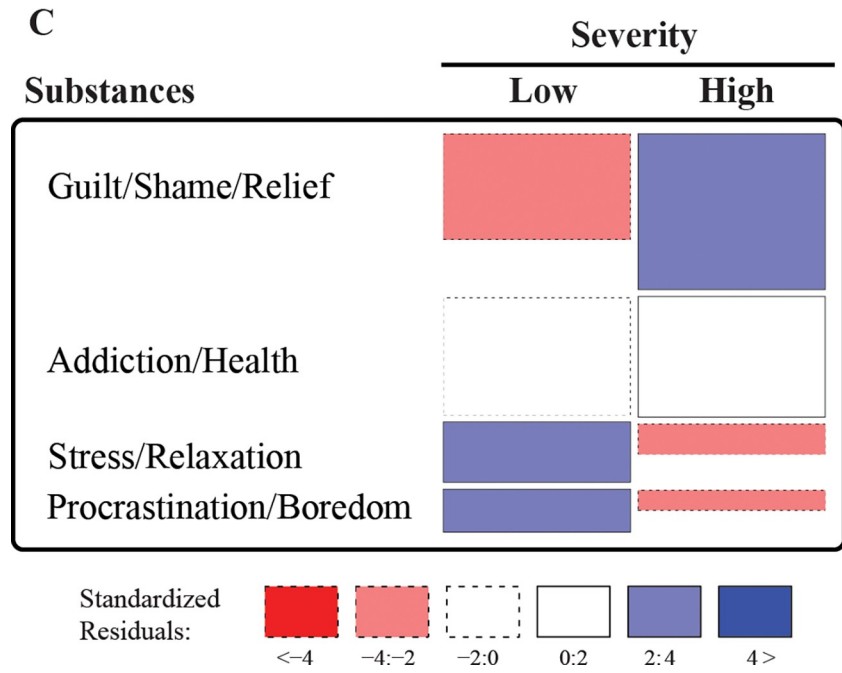

**Fig 3.** Graphical representation of differences in MRs between (A) psychoactive substance uses and potentially addictive behaviors, (B) low-intensity and high-intensity potentially addictive behaviors, and (C) low-intensity and high-intensity psychoactive substance uses. Colors indicate how far the observed frequencies deviate from what would be expected under the assumption of independence between the variables.

representations not limited to the psychoactive substance or potentially addictive behavior. Guilt has been described previously as a primary affective response in case of alcohol and other substance use disorders [41], tobacco use [42], problematic social networking site use [43] and compulsive sexual behavior and problematic pornography use [44–48]. Both "shame" and "guilt" were frequent associations, suggesting that the suspected view upon the self of others and self-reviewing and self-judgment can be crucial mechanisms [49], either restricted to the addiction–guilt -, or generalized to self-identity–shame–[41], but not differentiated strictly among individuals' MRs.

*Addiction/Health* (21%): Since frequent associations of this representation were "addiction", "pain", "harmful", and "unhealthy" (i.e., concepts belonging to health), it can be hypothesized that the concept of addiction and health risks are closely connected. The dominating emotion of this module was worries, presumably about health-related issues, in line with the extensive literature on health-related harms and substance addiction (e.g., [50]).

*Procrastination/Boredom* (29%): Frequent associations within this representation were "waste of time", "unnecessary", "boredom", "constraints" and "fatigue", all of which reflects passive acceptance rather than an active involvement. Previous research has described a possible causal link between leisure-boredom and substance abuse [51] and different behavioral addictions (internet: [52,53], sex: [54], pornography: [55,56], and smartphone use and compulsive shopping: [57]).

*Stress/Relaxation* (15%): Frequent associations were dominantly positive within this representation; "calm", "recreation", "joy" and "good" with only one negative (i.e., "stress"). The dominating emotions were also positive (e.g., joy, relaxation) but negative emotions were also present (e.g., worries, anxiety), although with lower likelihood than in the other three representations (see Fig 2A and Table 3). Relaxation, as a motivational factor has been described in case of many potentially addictive behaviors or psychoactive substance use, such as in case of alcohol use [58], tobacco use [59], cannabis use [60], overeating [61], gaming [62,63], hedonic shopping [64], problematic internet use [65], pornography use [55,56], and social media use [66].

Comparing the MRs with psychoactive substance uses and potentially addictive behaviors, a number of important differences were observed. Importantly, the distribution of Addiction/Health and Procrastination/Boredom MRs were different across substance uses and problem behaviors (see Fig 3A), indicating meaningful differences in the underlying cognitive evaluation processes. In case of behavioral addictions, one crucial measure of problem severity is the time spent on the activity [67]. Since there is no substance involved, tolerance is also reflected in the amount of time engaged with the activity, which can lead to the significant overpathologization of everyday life activities [9,68]. Based on the present study's results, it is proposed that it is not the time spent on the behavior *per se* that is problematic, but what the individual judges to be the time wasted from it, which might be a better indicator of the severity of behavior-related problems. This also supports Griffiths' [12] contention that time (in and of itself) should not be used as a criterion to assess behavioral addictions and that it is how the time conflicts with other important activities that is key. To illustrate with the aforementioned guitar playing example [9], assessing solely the time allocated to guitar playing is not sufficient to determine tolerance/addiction. However, if completed with the assessment of perceived wasted time, high scores will possibly reflect the level of tolerance. This reasoning is supported

by the fact that Time-wasting MRs were more frequent among individuals reporting high severity involvement in a behavior compared to low severity. While time wasting activities are not explicitly included in the diagnostic criteria of the ICD-11 [69], excessive time spent on a behavior that interferes with other important areas of life may be related it. For Gaming Disorder, the ICD-11 criteria state that the gaming behavior "takes precedence over other life interests and daily activities" and that the individual continues to engage in gaming despite "the occurrence of negative consequences", which might also lead to the percept of wasted time. For Gambling Disorder "gambling is often accompanied by a preoccupation with the activity" that can interfere with other important areas of life. Similarly, while the I-PACE model doesn't explicitly mention time wasting as a criterion, it does recognize that excessive engagement in certain activities (such as gaming, internet use, or gambling) can lead to negative consequences and interfere with daily functioning [70]. Thus, time wasting could be considered as a consequence or outcome of addictive behaviors rather than a standalone criterion within both the ICD-11 and I-PACE approach.

Participants with potentially addictive substance use habits mentioned associations from the Addiction/Health MR more frequently, indicating that individuals tend to classify their "disturbing excessive activity" as addiction if a substance is involved. This is in line with previous findings that non-professional participants perceived activities involving drug use (e.g. alcohol, smoking) to be more addictive than those that do not (e.g. exercise, watching TV) [71]. This might reflect the long-held scientific viewpoint–although this is now changing with the inclusion of behavioral addictions into the *Diagnostic and Statistical Manual of Mental Disorders* [72] and *International Classification of Diseases* [69]. This is also in line with the rich scientific literature on the health consequences of substance use addictions (e.g., [50,73]), and that the awareness of health hazards related to addictive substances is relatively high (e.g., [74,75]). Another factor that may contribute to the observed difference is that behavioral addictions arguably have a less direct negative or immediate impact on health (e.g., excessive internet use does not directly affect health, but can contribute to a decrease in physical activity, which can lead to a negative assessment of physical health [76]).

Guilt/Shame/Relief was an equally important MR for both substance misuse and potentially addictive behaviors, suggesting that self-criticism and evaluation [49] are determinative processes irrespective of the focus of the substance or behavior. However, importantly, the dynamics of the underlying cognitive-psychological processes might be different. For individuals with high-intensity substance use, Guilt/Shame/Relief was more frequent compared to low-intensity substance use, and Stress/Relaxation and Procrastination/Boredom were more frequent for low-intensity substance use. In other words, while in case of low-intensity use, all four representations were present, in case of high-intensity use, Stress/Relaxation and Procrastination/Boredom were highly diminished, and Guilt/Shame was the dominant representation. Although the current version of the International Classification of Diseases (ICD-11) primarily focus on the behavioral and physical symptoms of substance use—such as impaired control, social impairment, and pharmacological criteria—negative emotional states that can be associated with substance use disorders are also included [69], thus guilt and shame may be considered as part of it. Similarly, guilt and shame are not explicitly mentioned in the I-PACE model, some of the elements of the model may indirectly address these emotions. For example, the cognitive component of the model includes beliefs and expectations related to substance use which may include negative self-evaluations and self-blame [70], which can lead to feelings of guilt and shame.

More frequent associations from the Guilt/Shame/Relief MR in case of high severity substance use is consistent with cyclical models of addiction that expect more problematic forms of substance use to be associated with higher levels of shame [77]. A possible explanation for

**Table 4. Participants' characteristics according to mental representation membership.**

|  | Guilt/Shame/Relief | Addiction/Health | Stress/Relaxation | Procrastination/Boredom | p |
|---|---|---|---|---|---|
| n | 229 | 138 | 74 | 180 |  |
| Age, mean ±SD | 35 ± 10.7 | 37.1 ± 12.0 | 34.9 ± 10.5 | 33.4 ± 12.0 | - |
| Women (%) | 68.9% | 71.0% | 71.6% | 73.8% | - |
| High intensity (%) | 66.3% | 74.2% | 60.0% | 60.4% | - |
| Substance (%) | 37.7% | 36.6% | 15.0% | 10.5% | < .001 |
| Behavior (%) | 35.3% | 16.7% | 10.2% | 37.7% |  |
| Most frequently mentioned associations |  |  |  |  |  |
| 1st | Guilt (88) | Addiction (64) | Calm (53) | Time wasting (123) |  |
| 2nd | Shame (55) | Pain (61) | Recreation (37) | Unnecessary (76) |  |
| 3rd | Weakness (31) | Harmful (60) | Joy (33) | Boredom (76) |  |
| 4th | Anxiety (30) | Expensive (54) | Good (26) | Constraints (43) |  |
| 5th | Frustration (29) | Unhealthy (47) | Stress (25) | Fatigue (42) |  |

the observed asymmetry between psychoactive substances and potentially addictive behaviors regarding guilt/shame as a function of problem intensity might be that guilt is an essentially social phenomenon [78], therefore it is probably more influenced by the highly stigmatized state of substance use (e.g., see [31]). It is important to note that the Stress/Relaxation MR decreased for both high-intensity behaviors and substance use, consistent with the predictions of Incentive Sensitization Theory (IST, [79]) that the proportion of positive feelings decreases with the development of addiction (for various problematic behaviors, such as gaming, excessive shopping and social media use [80]). Relaxation was the least frequent MR (see Table 4), with no difference between psychoactive substances and potentially addictive behaviors. The fact that only one–the rarest–MR reflected positive aspects is also in line with the assumptions of IST that addiction is usually not driven by positive emotions (e.g., [79]).

The present study provides a comprehensive structure of *"disturbing excessive activity"*-related expressions, improving the understanding of individuals' representations and emotions. The results indicated that a different focus of intervention might be required based on problem behavior type (substance use/non-substance-related potentially addictive behaviors) and the intensity of the problem. In the case of substance-related behaviors, a shift is present in the MRs from the physical (Addiction/Health) to the psychological (Guilt/Shame/Relief) harms. Since shame [81] and stigma [82] have been identified as a major factor for relapse and the failure to seek help [83], the results of the present study support that shame management is an important therapeutic step (e.g., [77,84,85]), especially in case of intensive substance users. Also, the associations and linked emotions of the Addiction/Health representation raises the possibility that health anxiety might be a comorbid condition of problematic use and addiction. This would not only contribute to personal suffering [86], but individuals may be at an increased risk of engaging in substance use when experiencing obsessive health concerns [87]. Consequently, its relationship to addiction should be further investigated. On the other hand, for high-intensity behaviors, Procrastination/Boredom was the most frequent MR. Therefore, interventions to improve time management skills might be a promising in reducing high-intensity engagement in different potential behavioral addictions.

Wang [53] reported that free-time management skills were protective against leisure boredom, which significantly predicted problematic internet use. Boredom has been reported to be a significant predictor of other problematic behaviors, such as pornography use [56], smartphone use [88], and internet gaming [89]. Also, Altiner et al. [90] reported a negative relationship between time management skills and the level of problematic internet use. At first glance,

it may seem counter-intuitive that the Addiction/Health representation was less frequent for high-intensity compared to low intensity behaviors. This is partly explained by the fact that the frequency of Procrastination/Boredomrepresentation was higher, so the relative frequency of Addiction/Health necessarily decreased (i.e., an intense problematic behavior is more likely to trigger the concept of procrastination and boredomrelative to addiction and health). Another contributing factor might be that since quantitative norms for behaviors are harder to establish for individual or social levels than for psychoactive substance use, perceptions of being addicted to non-substance behaviors may be more driven by personal attitudes [91], making moral incongruence a relevant factor in the evaluation of potentially addictive behaviors (e.g., [92,93]).

A number of limitations to the present study warrant consideration. The data were all self-report and the sample was self-selected, both of which may have introduced methods biases (e.g., over-reporting or under-reporting, volunteer effects), potentially contributing to the high proportion of female participants. Despite the preprocessing efforts, which involved merging and translating the FWAs by two separate coders, the translations do not adequately account for the cultural disparities between the Hungarian and English languages. As a result, there may be minor inaccuracies in the outcomes. However, the primary objective of this analysis was not centered around individual FWAs, but rather on interpreting the collective FWAs within the modules, with the intention of minimizing inconsequential variations in translation. The present study did not test MRs and emotions explicitly across individuals with a clinical diagnosis of substance use disorder or behavioral addiction. Rather, potentially problematic use was classified as low-intensity or high-intensity based on a single item from the Screener for Substance and Behavioral Addictions [33]. Also, the data are not suitable to test more detailed associations between perceived wasted time and addiction-related measures (e.g., craving, motivation to quit, or subjective harm). Therefore, further studies are necessary to corroborate the assumption regarding perceived wasted time as a possible criterion when assessing for behavioral addictions.

Future studies should also consider the application of a longitudinal design, assessing use/behavior-related associations across multiple time-points. As the results of the present study suggest, changes in mental representations are to be expected, and exploring their temporal dynamics would be useful to better understand the emergence and evolution of maladaptive (addictive) behaviors. In addition, examining the temporal dynamics of mental representations may highlight important differences and similarities between the development of substance addictions and behavioral addictions, which may contribute to the theoretical basis for effective health communication and interventions. Further, to elaborate on the identified mental representations, qualitative research methods such as interviews and focus groups would be beneficial in gaining a more detailed understanding.

## Conclusion

The free word association approach combined with a data-driven co-occurrence network analysis is a promising technique in mapping MRs and feelings toward potentially addictive substances and behaviors. The distribution of Addiction/Health and Procrastination/Boredom mental representations varied significantly between substance use and other addictive behaviors, suggesting that different cognitive evaluation processes underlie them. The Addiction/Health representation was found to be more prevalent for substances, while the Procrastination/Boredom representation was more frequent for other addictive behaviors, and its frequency increased with the self-reported intensity of the behavior. Guilt/Shame/Relief was equally common for both substances and behaviors, but its likelihood increased with the

intensity of substance use. These differences in the mental representations of substances and behaviors may have implications for the development of more accurate diagnostic components for potential behavioral addictions and could help address ongoing debates in the scientific literature, as noted by Billieux et al. (2015b).The common overlaps of representations for both psychoactive substances and potentially addictive behaviors further provides support for the scientific viewpoint that addiction may develop and be present even in the absence of psychoactive substances.

## Supporting information

**S1 Appendix.**
(XLSX)

**S2 Appendix.**
(DOCX)

## Author Contributions

**Conceptualization:** Domonkos File, Bálint File, Zsolt Demetrovics.

**Data curation:** Domonkos File.

**Formal analysis:** Domonkos File, Bálint File.

**Funding acquisition:** Domonkos File, Zsolt Demetrovics.

**Investigation:** Domonkos File.

**Methodology:** Domonkos File, Bálint File.

**Project administration:** Domonkos File.

**Supervision:** Zsolt Demetrovics.

**Validation:** Beáta Bőthe, Mark D. Griffiths, Zsolt Demetrovics.

**Visualization:** Domonkos File, Bálint File.

**Writing – original draft:** Domonkos File, Bálint File.

**Writing – review & editing:** Domonkos File, Beáta Bőthe, Mark D. Griffiths, Zsolt Demetrovics.

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
