## [Decision Letter · Decision Letter 0]

10 Apr 2023

PONE-D-23-04935Investigating mental representations of psychoactive substance use and other potentially addictive behaviors using a data driven network-based clustering methodPLOS ONE

Dear Dr. File,

Please accept my apologies for the delayed feedback (that happened due to a protracted search for available reviewers), along with the assurances of our high consideration for your submission to the Plos ONE journal.

I am writing to inform you that your manuscript (ID: PONE-D-23-04935) has been thoroughly evaluated by one competent reviewer and myself, and happy to confirm that both evaluations conclude with a recommendation for minor revision. So, after careful consideration, we feel that your work has merit but does not fully meet PLOS ONE’s publication criteria as it currently stands. Namely, the minor revision suggests that the required changes are not considered major (labor-intensive or time-consuming), but they are still required in order to proceed in the next stage of the review process. Namely, you will see that, both evaluations raise concerns as regards the methodological approach (related to the selection of data, exclusion of participants, translation praocedures etc), and the open-science practices (related to availability of the data and pre-registration of the study). Hence, they must be addressed in a very elaborate manner, both in your response and inside the revised version of the manuscript.

That being said, the study is deemed as interesting and potentially beneficial for both the public and the experts because: a) it shifts the focus on the public and draws conclusions directly from their subjective reports, emphasizing their experiences as a way to ground theoretical considerations and allowing them to become so-creators of those theoretical considerations; b) it offers fairly new approach (perspective) for qualifying/quantifying of potentially addictive substances and behaviors.

If you agree to revise your manuscript please be mindful to submit by May 25 2023 11:59PM. If you will need more time than this to complete your revisions, please reply to this message or contact the journal office at plosone@plos.org. Upon submitting your revised manuscript, please make sure meet the journal requirements and include the following items:

With this I conclude my letter, sincerely hoping that it would help advance this interesting line of inquiry.

We look forward to receiving your revised manuscript.

With warm regards,

Biljana Gjoneska

Academic Editor

PLOS ONE

Journal Requirements:

"The study was supported by the National Research, Development and Innovation Office (K126835, K131635, PD138976). BB was supported by a postdoctoral fellowship from the SCOUP Team – Sexuality and Couples – Fonds de recherche du Québec, Société et Culture."

"ELTE Eötvös Loránd University receives funding from the Szerencsejáték Ltd. to maintain a telephone helpline service for problematic gambling. ZD has also been involved in research on responsible gambling funded by Szerencsejáték Ltd. and the Gambling Supervision Board and provided educational materials for the Szerencsejáték Ltd’s responsible gambling program. The University of Gibraltar receives funding from the Gibraltar Gambling Care Foundation. MDG’s university has received research funding from Norsk Tipping (the gambling operator owned by the Norwegian Government). MDG has also received funding for a number of research projects in the area of gambling education for young people, social responsibility in gambling and gambling treatment from Gamble Aware (formerly the Responsible Gambling Trust), a charitable body which funds its research program based on donations from the gambling industry. MDG regularly undertakes consultancy for various gaming companies in the area of social responsibility in gambling. However, these funding sources are not related to the present study and the funding institution had no role in the study design or the collection, analysis, and interpretation of the data, writing the manuscript, or the decision to submit the paper for publication. "

Additional Editor Comments:

Below please refer to the detailed reviews for better insight on the raised issues, concerns, comments and suggestions:

I. Major issues

• Research practices:

◦ Data availability statements: In your submission form you have first indicated that the “Data are fully available with no restriction”, and then noted that the “Data of this study are available from the corresponding author upon request”. Please note that both statements are conflicting with each other and with Plos ONE policies. The journal places utmost importance on transparency and open-science practices that increase the visibility, trust, and reuse of research, so simply “stating that data will be made available upon request is not sufficient” (as clearly instructed in the submission form). In such case you should make effort to “explain your exceptional situation” (as per journal’s guidelines). So, I would advise that you make sure to either a) make your data publicly available (especially during the next round of revisions so as to enable critical evaluation of your work); or b) offer strong argumentation for your decision to sustain from doing so.

◦ Self-referencing instances: On couple of places throughout the manuscript, a single reference (to an earlier work by the lead author) is provided in support of the statements regarding some of the methodological aspects (on page 3, 5 and 8). So, I advice the authors to trace other instances that are related to similar methodological approaches (which do not necessarily have to be related to the same subject matter) and include them correspondingly/

• Methodological approaches:

◦ Using a qualifier for a problem ("excessive activity") to refer to the problem itself ("addiction") seems to be a potential limitation of this study (albeit reasonable and justifiable, as already explained by the authors). Namely, it might be a reason why: a) only a portion of the entire list was retrieved and included in a preliminary list with 1374 unique associations (presuming that the initial poll was much bigger since each of those 621 participants was asked to provide 5 associations); and b) a big portion of the final list with 254 associations was eliminated as idiosyncratic and not considered for further analysis (i.e., 177 free-word associations). First, the qualifier might be regarded as equally abstract and ambiguous, triggering associations that fall outside the scope of the study (e.g., excessive activity can also signify extravagant, eccentric, abundant and unconventional behaviors). Second, the qualifier is not specific enough to pertain exclusively to the problem of addiction. In addition, the posed question is emotionally loaded because it implies negative emotions (since it asks participants to imagine excessive activity that was most bothering for them). So, in a way, it could affect participants’ responses on the PANAS scale. In summary, I would encourage the authors to: a) acknowledge this procedural constrain as a potential limitation of this study; and b) provide more details as regards the idiosyncratic associations that were eliminated from further analysis (considering the fact that they comprise 2/3 of the final FWA list).

◦ It remains unclear why authors decided to translate participants’ responses into English? Was it due to some methodological constrains (related to the tool for FWA analysis)? In any case this should be: a) justified inside the manuscript; and b) acknowledged as one of the potential limitations of this study. A culturally-adapted translation requires extensive work (coordinating work of language and subject experts), to make sure that important data is not lost in the process of translation, so this should be properly acknowledged inside the limitation section.

• Presentation of results:

At present, the overall tabular and graphic presentation of results appears a bit raw and undeveloped. For instance, I would suggest including a new table to offer an overview on socio-demographic characteristics of the population (as more effective variation to the narrative summary). Also, Table 1 could be further improved to include a caption and a legend, as well descriptive statistics related to the problem intensity or severity of use (i.e., responses to the statement “I did it too much in the past 12 months”) for each of the of the response categories (i.e., each of the listed addictive substances or behaviors) including: mean age, gander ratio, mean value and standard deviation per response category, as well as response frequency. Likewise, a legend is missing in all of the provided figures. For instance, it would be good to spell-out the choice of the red/white/blue color in Figure 3. Also, all figures are low-resolution so a better-quality images should be included in the next round.

• Research implications:

The authors were mindful to discuss the implications of their work as regards future therapeutic approaches. However, I invite the authors to comment on their findings in relation to: a) existing frameworks of addictive behaviors (in particular I-PACE model by Brand et al, 2016); b) diagnostic criteria and measures of addictive behaviors (especially in reference to ICD-11). In particular, it might be useful to understand whether the identified emotions (guilt/shame) and behaviors (tame-wasting activities) are already emphasized in the existing diagnostic criteria, definitions and frameworks of addictive behaviors (and if so, in what way)?

II. Minor issues:

• Sampling procedures:

Authors provide info as regards the data collection procedures, but not as regards the sampling procedure. From what I gather, the authors relied on convenience sampling (or maybe a combination of convenience and community sampling), but this should be clearly outlined. Also, in case the obtained sample is representative of Hungarian population in terms of age (or any other socio-demographic parameter), please make sure to report it accordingly.

• Exclusion criteria:

In the Results section, the authors mention that participants who did not provide specific response to the posed question as regards FWA (i.e., have selected the option “other”) were excluded from further analysis. However, the authors are also encouraged to provide an overview of the entire exclusion criteria in the Method section (on page 4 of the manuscript). Furthermore, criteria for data cleaning are already outlined (excluding free-word associations which were mentioned by less than 10 participants, and PANAS labels which were provided by less than 50% of the participants), but the authors do not clarify if the applied criteria for data cleaning pertained to all responses from a single participant (and if yes, why), or only to the particular response in question.

• Ethics procedures:

A statement regarding the informed consent and the anonymity of obtained data appears twice (on page 3 and 4) under the ethics procedures of the manuscript. Please make sure to avoid redundancy across all text.

• FWA methodology:

Some of the concluding paragraphs in the Introductory section (pertaining to the FWA) belong to the Method section (Analyses subsection) of the manuscript, so should be placed accordingly. In addition, I encourage the authors to include a mention of the tool for FWA analyses (operative software and statistical package along with the accompanying references), as a way to avoid criticism of using potentially unreliable tool (since there are many free FWA generators on Internet).

• The item from the Screener for Substance and Behavioral Addictions (SSBA):

It might not be proper to mention psychometric properties of the instrument, since the authors have borrowed a single item from that instrument and do not perform psychometric evaluation of that instrument.

• Labeling of the mental representations:

At present, it seems that a unified approach in choosing labels is lacking because: a) some of the labels emphasize the emotional states (e.g., guilt/shame), while others focus on the behavioral aspects (e.g., time-wasting); b) some of the labels emphasize the underlying dichotomy (e.g., addiction/health) while other focus solely on the negative aspects (e.g., guilt/shame) or positive aspects (e.g., relaxation). This may confuse the readers, so it might be useful to choose an approach that seems more unified when creating labels (but this suggestion is optional). For instance, “time-wasting” could be rephrased to say “time-wasting perception” or replaced with “boredom” (as most frequently reported emotion in relation to time-wasting), in case the authors opt to emphasize the emotional aspects. Also, if authors choose to emphasize the underlying dichotomy they might consider some of the following combinations (in addition to “Addiction/Health”); “Stress/Relaxation”, “Guilt/Shame/Relief”, “Procrastination/Boredom” etc. However, I understand that the labels stem from the FWA network, so choosing different labels might be not entirely accurate, so please consider this suggestion is optional, but do offer your reasoning as regards your choices in your response letter.

• Conclusions:

The conclusion section serves to reiterate the main findings, so briefly listing the main differences and similarities (at appropriate places in that section) would be very useful as it would highlight the take-home messages for the readership audiences.

Reviewers' comments:

Reviewer's Responses to Questions

**Comments to the Author**

1. Is the manuscript technically sound, and do the data support the conclusions?

Reviewer #1: Yes

2. Has the statistical analysis been performed appropriately and rigorously? 

Reviewer #1: Yes

3. Have the authors made all data underlying the findings in their manuscript fully available?

Reviewer #1: No

4. Is the manuscript presented in an intelligible fashion and written in standard English?

Reviewer #1: Yes

5. Review Comments to the Author

Reviewer #1: Thank you for reviewing this manuscript and learning about mental representations (MR) of different addictions using a data driven network-based clustering method.

Guilt-Shame was a frequent MR, suggesting generalized representations not limited to psychoactive substances or potentially addictive behaviour. Addiction-Health was another frequent MR, with the main emotion being worries, presumably about health-related issues. Time-wasting and Relaxation were also observed, with positive emotions like joy and relaxation being present in the latter. Addiction-Health and Time-wasting were differently present in psychoactive substance use and behavioural addictions. Addiction-Health was more dominant with substance use.

While all four MRs were present for low-intensity substance use, high-intensity use was associated with Guilt-Shame, Relaxation and Time-wasting were present in low intensity substance use.

The study found guilt-shame was an important motivation for both substance misuse and addictive behaviours. For high-intensity use, guilt-shame was dominant with relaxation and time-wasting highly diminished. Cyclical models of addiction show shame is associated with more problematic forms of substance use.

This study presents a comprehensive and well-thought out research design. The authors made a deliberate decision to use “excessive activity” instead of addiction as a call word in order to avoid triggering common knowledge and to minimize stigmas. Furthermore, the authors were also careful to use the Positive and Negative Affect Scale (PANAS) labels to better interpret the results. The use of the Screener for Substance and Behavioral Addictions (SSBA) was also beneficial for the assessment of potentially addictive behaviours. The authors also used an extended form of modularity to analyse the co-occurrence of words and phrases in the network.

Notes

The use of free association tests and network calculations in this study is an interesting and innovative approach for exploring mental representations of addiction. However, there are some areas in which this approach could be improved.

I'm concerned about data filtering and processing methods as these seem, to an extent, arbitrary, and could result in inflated results. Why were association words with a lower than 10 removed? Why this number? Why were emotion labels with mentions under 50% removed? Why this number? Did one person translate terms to English or several? Did the authors check interrater reliability? What does it mean that words were merged if they were close synonyms? Was this principled in some way? Since there was no preregistration, all these analytical decisions can be seen as post hoc and they can result in inflated effects in the results.

I have concerns about a lack of a discernible control group for the data. Have the authors considered using either word vectors, like word2vecor, or large language models, such as BERT or GPT3 to measure distances between the associative terms and the emotional labels? This could provide a neutral baseline, where subject addictions are not present.

The problem with the design is that it relies heavily on subjective responses from participants. This could lead to difficulties in accurately interpreting and analysing the results, as different participants may interpret the words or emotions differently. Another point is raised that the word association may reflect to the linguistic environment where participants live (and may not reflect their subjective experiences).

Additionally, the use of only one item from the Screener for Substance and Behavioral Addictions (SSBA) may not provide an accurate measure of problem severity. Considering subjective responses, qualitative research methods such as interviews and focus groups could be used to gain a better understanding of participants' experiences with addiction. This could provide further insight into the mental representations of addiction, as well as potential underlying causes and motivations.

I suggest that the manuscript be published subsequent to the receipt of the authors' responses.

6. PLOS authors have the option to publish the peer review history of their article (what does this mean?). If published, this will include your full peer review and any attached files.

Reviewer #1: **Yes: **Jozsef Racz

---

## [Author Response · Author response to Decision Letter 0]

23 May 2023

We would like to extend our sincere gratitude to the reviewers for their valuable and insightful comments on our manuscript. Their thorough review and constructive feedback have immensely contributed to improving the quality and clarity of our work. We deeply appreciate the time and effort they dedicated to carefully evaluating our research and providing thoughtful suggestions for enhancement.

Reviewer 1

Self-referencing instances: On couple of places throughout the manuscript, a single reference (to an earlier work by the lead author) is provided in support of the statements regarding some of the methodological aspects (on page 3, 5 and 8). So, I advice the authors to trace other instances that are related to similar methodological approaches (which do not necessarily have to be related to the same subject matter) and include them correspondingly.

To address this concern, we have included a new paragraph in the introduction:

“In spite of the availability of several clustering solutions, such as factor analytic methods (Szalay, L. B., & Deese, J. (1978). Subjective meaning and culture: An assessment through word associations. Hillsdale: Erlbaum) and hierarchical agglomerative clustering (Popoola, I. O., Anders, S., Feuereisen, M. M., Savarese, M., & Wismer, W. V. (2021). Free word association perceptions of red meats; beef is ‘yummy’, bison is ‘lean game meat’, horse is ‘off limits’. Food Research International, 148, 110608), the current study implemented a modular analysis of the co-occurrence network of free word associations (FWAs) (File et al., 2019). We applied this approach, as our methodological research has demonstrated the capability of modular analysis in establishing a reproducible result (File, 2019) and connecting mental representations (MRs) to cue-specific behaviours and attitudes of participants (File et al., 2019; Gero et al., 2020).”

• Methodological approaches: Using a qualifier for a problem ("excessive activity") to refer to the problem itself ("addiction") seems to be a potential limitation of this study (albeit reasonable and justifiable, as already explained by the authors). Namely, it might be a reason why: a) only a portion of the entire list was retrieved and included in a preliminary list with 1374 unique associations (presuming that the initial poll was much bigger since each of those 621 participants was asked to provide 5 associations); and b) a big portion of the final list with 254 associations was eliminated as idiosyncratic and not considered for further analysis (i.e., 177 free-word associations). First, the qualifier might be regarded as equally abstract and ambiguous, triggering associations that fall outside the scope of the study (e.g., excessive activity can also signify extravagant, eccentric, abundant and unconventional behaviors). Second, the qualifier is not specific enough to pertain exclusively to the problem of addiction. In addition, the posed question is emotionally loaded because it implies negative emotions (since it asks participants to imagine excessive activity that was most bothering for them). [konkrét számok, hogy… true etc. the scope of the study was to map the general representation of the investigated substances, and the current approach and sample is not yet suitable to explora such nuiances, although would be highly interesting. Fuck off, xxx] So, in a way, it could affect participants’ responses on the PANAS scale. In summary, I would encourage the authors to: a) acknowledge this procedural constrain as a potential limitation of this study; and b) provide more details as regards the idiosyncratic associations that were eliminated from further analysis (considering the fact that they comprise 2/3 of the final FWA list).

◦ It remains unclear why authors decided to translate participants’ responses into English? Was it due to some methodological constrains (related to the tool for FWA analysis)? In any case this should be: a) justified inside the manuscript; and b) acknowledged as one of the potential limitations of this study. A culturally-adapted translation requires extensive work (coordinating work of language and subject experts), to make sure that important data is not lost in the process of translation, so this should be properly acknowledged inside the limitation section.

Word and association networks exhibit scale-free distributions (Keczer, Hero), which can be interpreted as Zipf's law from a frequentialist perspective. In other words, there are a few frequently mentioned free word associations (FWAs), while many FWAs have only a few mentions. In our study, the initial list of 1,374 unique associations indicates that the 1,374 different FWAs were generated from 661 responses, each containing five answers. After spell-checking and merging synonyms, this number decreased significantly. For instance, in the Hungarian language, the use of accents is often inconsistent, particularly in smartphone responses, leading to distinct FWAs in the preliminary list.

From a theoretical standpoint, we excluded rare associations, as our goal was to represent the primary modules and disregard answers that likely do not form part of the stable social representation. These findings might be specific to the current sample and potentially not replicable. For example, one participant mentioned "shower," an intriguing individual result that might not appear in subsequent data collection. Methodologically, due to the low number of observations, idiosyncratic FWAs cannot provide stable information for linking to modules. We included a figure in the manuscript illustrating the total number of retained associations relative to the exclusion threshold. This figure demonstrates a breakpoint at a frequency of 10, indicating that excluding associations mentioned less than 10 times still retained 65% of the total FWAs in the analysis (image might not be displayed in the browser, please see the docx uploaded Response to Reviewers):

Figure Supplementary. Relationship between the frequency of FWAs and the percentage of excluded FWAs. The data exhibits a breakpoint at a frequency of 10, as indicated by a linear fit to the function.

For detailed presentation of results, associations were translated into English. Figure 1 displays the modular membership of each association, and translating the answers into English likely aids reader comprehension. FWAs recorded in non-English-speaking countries are often translated by other authors as well (e.g., see Ernst‐Vintila, A., Delouvée, S., & Roland‐Lévy, C. (2011). Under threat. Lay thinking about terrorism and the three‐dimensional model of personal involvement: a social psychological analysis. Journal of Risk Research, 14(3), 297-324; De Deyne, S., & Storms, G. (2008). Word associations: Norms for 1,424 Dutch words in a continuous task. Behavior research methods, 40(1), 198-205.), however it is certainly a valuable advice that a culturally-adapted translation would increase the relevance of the results. 

We added these thoughts to the limitations:

Despite the preprocessing efforts, which involved merging and translating the FWAs by two separate coders, the translations do not adequately account for the cultural disparities between the Hungarian and English languages. As a result, there may be minor inaccuracies in the outcomes. However, the primary objective of this analysis was not centered around individual FWAs, but rather on interpreting the collective FWAs within the modules, with the intention of minimizing inconsequential variations in translation

• Presentation of results:

At present, the overall tabular and graphic presentation of results appears a bit raw and undeveloped. For instance, I would suggest including a new table to offer an overview on socio-demographic characteristics of the population (as more effective variation to the narrative summary). Also, Table 1 could be further improved to include a caption and a legend, as well descriptive statistics related to the problem intensity or severity of use (i.e., responses to the statement “I did it too much in the past 12 months”) for each of the of the response categories (i.e., each of the listed addictive substances or behaviors) including: mean age, gander ratio, mean value and standard deviation per response category, as well as response frequency. Likewise, a legend is missing in all of the provided figures. For instance, it would be good to spell-out the choice of the red/white/blue color in Figure 3. Also, all figures are low-resolution so a better-quality images should be included in the next round.

We are grateful for the suggestions provided, and as a result, we have included a new table that provides an overview of the demographic characteristics of the sample. Additionally, we have extended Table 1 (now Table 2) to include more information. Also, high resolution images were attached and the legend of figure 3 was completed with the following sentence: “Colors indicate how far the observed frequencies deviate from what would be expected under the assumption of independence between the variables.”

Research implications:

The authors were mindful to discuss the implications of their work as regards future therapeutic approaches. However, I invite the authors to comment on their findings in relation to: a) existing frameworks of addictive behaviors (in particular I-PACE model by Brand et al, 2016); b) diagnostic criteria and measures of addictive behaviors (especially in reference to ICD-11). In particular, it might be useful to understand whether the identified emotions (guilt/shame) and behaviors (tame-wasting activities) are already emphasized in the existing diagnostic criteria, definitions and frameworks of addictive behaviors (and if so, in what way)?

We would like to thank the suggestion, the following parts were added: “While time wasting activities are not explicitly included in the diagnostic criteria of the ICD-11 (World Health Organization, 2018), excessive time spent on a behavior that interferes with other important areas of life may be related it. For Gaming Disorder, the ICD-11 criteria state that the gaming behavior "takes precedence over other life interests and daily activities" and that the individual continues to engage in gaming despite "the occurrence of negative consequences", which might also lead to the percept of wasted time. For Gambling Disorder "gambling is often accompanied by a preoccupation with the activity" that can interfere with other important areas of life. Similarly, while the I-PACE model doesn't explicitly mention time wasting as a criterion, it does recognize that excessive engagement in certain activities (such as gaming, internet use, or gambling) can lead to negative consequences and interfere with daily functioning (Brand et al., 2016). Thus, time wasting could be considered as a consequence or outcome of addictive behaviors rather than a standalone criterion within both the ICD-11 and I-PACE approach.” And “Although the current version of the International Classification of Diseases (ICD-11) primarily focus on the behavioral and physical symptoms of substance use - such as impaired control, social impairment, and pharmacological criteria - negative emotional states that can be associated with substance use disorders are also included (World Health Organization, 2018), thus guilt and shame may be considered as part of it. Similarly, guilt and shame are not explicitly mentioned in the I-PACE model, some of the elements of the model may indirectly address these emotions. For example, the cognitive component of the model includes beliefs and expectations related to substance use which may include negative self-evaluations and self-blame (Brand et al., 2016), which can lead to feelings of guilt and shame.”

II. Minor issues:

• Sampling procedures:

Authors provide info as regards the data collection procedures, but not as regards the sampling procedure. From what I gather, the authors relied on convenience sampling (or maybe a combination of convenience and community sampling), but this should be clearly outlined. Also, in case the obtained sample is representative of Hungarian population in terms of age (or any other socio-demographic parameter), please make sure to report it accordingly. 

It was clarified: “The samples were collected via online questionnaires, and survey completion took approximately 20 minutes. The sample used in this study may not be representative of the Hungarian population, as it was obtained through a popular Hungarian news portal and public, topic-irrelevant Facebook pages”

• Exclusion criteria:

In the Results section, the authors mention that participants who did not provide specific response to the posed question as regards FWA (i.e., have selected the option “other”) were excluded from further analysis. However, the authors are also encouraged to provide an overview of the entire exclusion criteria in the Method section (on page 4 of the manuscript). Furthermore, criteria for data cleaning are already outlined (excluding free-word associations which were mentioned by less than 10 participants, and PANAS labels which were provided by less than 50% of the participants), but the authors do not clarify if the applied criteria for data cleaning pertained to all responses from a single participant (and if yes, why), or only to the particular response in question.

We would like to thank for the comment. It has come to our attention that an important step of data cleaning was inadvertently omitted from the description. We have now included this step to ensure a more comprehensive and accurate representation of the process. “Free-word associations were (i) first spellchecked (transformed to lower case, removed accents, and manually corrected), (ii) lemmatized and (iii) translated to English. Associations were merged if their English translation was identical, or was a close synonym. While assessing the substance/behavior the participants thought of, 76 (out of 661) participants indicated ‘other’ from the list of choices (see Table 1). As it is not known whether their associations concerned a substance or a behavior, these participants were excluded from further analysis. FWAs mentioned by less than 10 participants were excluded from the analysis as such terms can refer to unstable or idiosyncratic parts of the representations (Sarrica, 2007, Abric, 1993) and a breakpoint at a frequency of 10 was observable (see Figure at Appendix), indicating that excluding associations mentioned less than 10 times still retained 65% of the total FWAs in the analysis. Participants whose associations' frequency for each category was lower than 10 were unable to be classified (n=40) and were subsequently excluded from subsequent analyses. This resulted in a final study cohort consisting of 546 participants. PANAS labels were considered relevant in the context of the study that were provided by at least 50% of participants overall. The decision to set this threshold was arbitrary, serving the purpose of maintaining stability in emotional labels and directing the study towards emotional states that, due to their frequent occurrence, were more likely to have a general consensus among the respondents. Appendix 2 contains the results pertaining to the labels that were not considered due to the implementation of this approach.

• Ethics procedures:

A statement regarding the informed consent and the anonymity of obtained data appears twice (on page 3 and 4) under the ethics procedures of the manuscript. Please make sure to avoid redundancy across all text. 

We are thankful for the comment, it was corrected. 

• FWA methodology:

Some of the concluding paragraphs in the Introductory section (pertaining to the FWA) belong to the Method section (Analyses subsection) of the manuscript, so should be placed accordingly. In addition, I encourage the authors to include a mention of the tool for FWA analyses (operative software and statistical package along with the accompanying references), as a way to avoid criticism of using potentially unreliable tool (since there are many free FWA generators on Internet).

We moved the explanation of the Free Word Association (FWA) procedure from the introduction to the methods section. 

We also emphasized that we employed an in-house developed tool with codes, which was validated in a methodological paper (File, 2019). In this methodological study, we demonstrated that the modular analysis can yield a stable structure (File, 2019) and that Mental Representations (MRs) can be associated with the cue-specific behaviours and attitudes of the participants (File et al., 2019; Gero et al., 2020).

Furthermore, we added the software applications used (Matlab and Gephi) and the related network toolbox (https://sites.google.com/site/bctnet/ (Rubinov & Sporns, 2010)) to the methods section.

• The item from the Screener for Substance and Behavioral Addictions (SSBA):

It might not be proper to mention psychometric properties of the instrument, since the authors have borrowed a single item from that instrument and do not perform psychometric evaluation of that instrument.

The sentence in question has been removed. 

Labeling of the mental representations:

At present, it seems that a unified approach in choosing labels is lacking because: a) some of the labels emphasize the emotional states (e.g., guilt/shame), while others focus on the behavioral aspects (e.g., time-wasting); b) some of the labels emphasize the underlying dichotomy (e.g., addiction/health) while other focus solely on the negative aspects (e.g., guilt/shame) or positive aspects (e.g., relaxation). This may confuse the readers, so it might be useful to choose an approach that seems more unified when creating labels (but this suggestion is optional). For instance, “time-wasting” could be rephrased to say “time-wasting perception” or replaced with “boredom” (as most frequently reported emotion in relation to time-wasting), in case the authors opt to emphasize the emotional aspects. Also, if authors choose to emphasize the underlying dichotomy they might consider some of the following combinations (in addition to “Addiction/Health”); “Stress/Relaxation”, “Guilt/Shame/Relief”, “Procrastination/Boredom” etc. However, I understand that the labels stem from the FWA network, so choosing different labels might be not entirely accurate, so please consider this suggestion is optional, but do offer your reasoning as regards your choices in your response letter.

We appreciate the suggestion to create a more unified approach in choosing labels. The initial idea using the current labels was to reflect the data-driven nature of the research: i.e., naming the module after the frequent associations. However, we do agree that the labels can be somewhat confusing for readers, and the suggested ones contribute to an easier understanding of the study, thus it was changed accordingly. 

• Conclusions:

The conclusion section serves to reiterate the main findings, so briefly listing the main differences and similarities (at appropriate places in that section) would be very useful as it would highlight the take-home messages for the readership audiences.

We would like to thank for the suggestion, the conclusion was extended with the following part: “The distribution of Addiction/Health and Procrastination/Boredom mental representations varied significantly between substance use and other addictive behaviors, suggesting that different cognitive evaluation processes underlie them. The Addiction/Health representation was found to be more prevalent for substances, while the Procrastination/Boredom representation was more frequent for other addictive behaviors, and its frequency increased with the self-reported intensity of the behavior. Guilt/Shame/Relief was equally common for both substances and behaviors, but its likelihood increased with the intensity of substance use. These differences in the mental representations of substances and behaviors may have implications for the development of more accurate diagnostic components for potential behavioral addictions and could help address ongoing debates in the scientific literature, as noted by Billieux et al. (2015b).”

Reviewer 2

I'm concerned about data filtering and processing methods as these seem, to an extent, arbitrary, and could result in inflated results. Why were association words with a lower than 10 removed? Why this number? 

From the theoretical point of view, we excluded rare associations as we aimed to present the major modules of the representation and ignoring answers, which probably not part of the stable social representation. It is important to note that these results may be specific to the particular sample used and may not be easily reproducible. For instance, a participant's mention of "shower" might be an intriguing individual result, but it might not reoccur in subsequent data collection.

From a methodological standpoint, due to the limited number of observations, idiosyncratic FWAs do not provide consistent information that can be reliably linked to the identified modules. To visually illustrate this, we included a figure in the manuscript Relationship between the frequency of FWAs and the percentage of excluded FWAs. The figure demonstrates that the function reaches a breakpoint at a frequency of 10. Furthermore, even when associations mentioned less than 10 times are excluded, it still retains 65% of the total number of FWAs included in the analysis.

Figure Supplementary. Relationship between the frequency of FWAs and the percentage of excluded FWAs. The data exhibits a breakpoint at a frequency of 10, as indicated by a linear fit to the function.

The following part was added to the manuscript: “FWAs mentioned by less than 10 participants were excluded from the analysis as such terms can refer to unstable or idiosyncratic parts of the representations (Sarrica, 2007, Abric, 1993) and a breakpoint at a frequency of 10 was observable (see Figure at Appendix), indicating that excluding associations mentioned less than 10 times still retained 65% of the total FWAs in the analysis.”

Why were emotion labels with mentions under 50% removed? Why this number? 

In our study, we sought to utilize a well-established measure of affective states, such as the Positive and Negative Affect Schedule (PANAS). Given that the analysis of Free Word Associations (FWAs) is exploratory in nature, we did not possess prior knowledge of the expected FWAs or the emotional states associated with them by the respondents. While we had some expectations, such as the likelihood of the emotion "guilt" being frequently mentioned, we recognized that the state of "alertness" might not be specific to the context of addiction. Consequently, we employed the complete PANAS scale and subsequently applied a similar frequency-based filtering approach as used in the FWAs. This approach ensured the stability of emotional labels and allowed us to focus on emotional states that, based on their high frequency, were likely to be consensual among the respondents.

We acknowledge the reviewer's comment regarding the use of the 50% threshold, recognizing that it is a heuristic value. To address this concern, we conducted statistical analyses for all the remaining 14 PANAS labels, and the results have been included as a supplementary table in the manuscript, and the following part was added to the manuscript: ‘The decision to set this threshold was arbitrary, serving the purpose of maintaining stability in emotional labels and directing the study towards emotional states that, due to their frequent occurrence, were more likely to have a general consensus among the respondents. Appendix 2 contains the results pertaining to the labels that were not considered due to the implementation of this approach.’ 

Did one person translate terms to English or several? Did the authors check interrater reliability? What does it mean that words were merged if they were close synonyms? Was this principled in some way? Since there was no preregistration, all these analytical decisions can be seen as post hoc and they can result in inflated effects in the results.

Two authors (D. File & B. File) were pre-processed the FWAs. The authors independently translated the FWAs and then discussed the inconsistent solutions. In the process, close synonyms were taken into consideration based on the English translations. Whenever two Hungarian words were translated into the same English word, they were merged. It is important to note that this merging rule specifically applied to synonyms. We acknowledge the limitations of the translation process, such as adopting the cultural differences of the English and Hungarian language. As a result, there may be minor inaccuracies in the outcomes. However, we would like to emphasize that the primary objective of this analysis was not focused on individual FWAs, but rather on interpreting the collective FWAs within the modules, which might minimize the inconsequential variations in translation and merging. We added these thoughts to the limitations.

I have concerns about a lack of a discernible control group for the data. Have the authors considered using either word vectors, like word2vecor, or large language models, such as BERT or GPT3 to measure distances between the associative terms and the emotional labels? This could provide a neutral baseline, where subject addictions are not present.

The problem with the design is that it relies heavily on subjective responses from participants. This could lead to difficulties in accurately interpreting and analysing the results, as different participants may interpret the words or emotions differently. Another point is raised that the word association may reflect to the linguistic environment where participants live (and may not reflect their subjective experiences).

Regarding the concerns that the word associations may reflect the linguistic environment where participants live, it is important to note that this is a potential limitation of any word association study. One the one hand FWA studies provides free responses, short, cheap data collection thus large respondent pools, but the lack of context compared to interviews can raise difficulties in the interpretation. Baseline conditions from language models would build on the large corpora. It assumes that FWAs and words in addictions related texts are in close relations. Here we would like to refer to the study Sandra Mollino. It was shown that there were fewer parallels, and far more differences, between corpus collocations and word association responses than has been previously assumed (https://www.degruyter.com/document/doi/10.1515/CLLT.2009.008/html). Considering that FWA differs from text word usage we assume that a proposed baseline combines two inseparable effects: the semantic information and the lack of addiction. Although the use of language models as a baseline is intriguing, we fully agree with the reviewer's suggestion that the FWA processing procedure should be further developed in future studies. As recommended, we could incorporate language models as a covariate in the model to account for the semantic aspect of associations. 

Additionally, we can consider utilizing established FWA databases like the University of South Florida Association Data Set (USF) by Nelson et al. (2004) or other large databases (https://link.springer.com/article/10.3758/s13428-018-1115-7). Although, these FWA databases are in English, it might help to create a baseline for focus-group related studies. Currently, we are acquiring a large database of psychologically/clinically interesting cues from thousands of respondents (e.g.: migrant, weight loss, COVID-19, addiction) and we plan to develop the technique on these cues.

In conclusion, we agree with the reviewer, that incorporating semantic and the corresponding sentiment information by language models could further emphasize the attitude relatedness of the association modules. Here, due to the absence of a baseline condition, our study focused on different subgroups of respondents based on their FWA modular memberships and examined differences in their perceptions and attitudes. Despite the semantic aspects of FWAs, our results inevitably reflect attitudes as well, as otherwise, the FWA modules would not demonstrate differences in attitudes.

Additionally, the use of only one item from the Screener for Substance and Behavioral Addictions (SSBA) may not provide an accurate measure of problem severity. Considering subjective responses, qualitative research methods such as interviews and focus groups could be used to gain a better understanding of participants' experiences with addiction. This could provide further insight into the mental representations of addiction, as well as potential underlying causes and motivations.

More detailed measures and assessments may be necessary to gain a better understanding of the nature and severity of addiction in the population being studied, which was included to the limitations: “Rather, potentially problematic use was classified as low-intensity or high-intensity based on a single item from the Screener for Substance and Behavioral Addictions (Schulter et al. 2018). Also, the data are not suitable to test more detailed associations between perceived wasted time and addiction-related measures (e.g., craving, motivation to quit, or subjective harm). Therefore, further studies are necessary to corroborate the assumption regarding perceived wasted time as a possible criterion when assessing for behavioral addictions. ” 

We greatly appreciate the suggestion to incorporate qualitative research methods such as interviews and focus groups to extend our insights on participants' experiences with addiction (using the presented FWA method to select topics and further elaborate on them with qualitative methods). However, we beleive that this suggestion may be out of the scope of the current study. Regarding this assumption, the following part was added to the manuscript: “Further, to elaborate on the identified mental representations, qualitative research methods such as interviews and focus groups would be beneficial in gaining a more detailed understanding.”

---

## [Decision Letter · Decision Letter 1]

7 Jun 2023

Investigating mental representations of psychoactive substance use and other potentially addictive behaviors using a data driven network-based clustering method

PONE-D-23-04935R1

Dear Dr. File,

We’re pleased to inform you that your manuscript has been judged scientifically suitable for publication and will be formally accepted for publication once it meets all outstanding technical requirements.

Kind regards,

Nicholas Aderinto Oluwaseyi

Academic Editor

PLOS ONE

Additional Editor Comments (optional):

Reviewers' comments:

Reviewer's Responses to Questions

**Comments to the Author**

1. If the authors have adequately addressed your comments raised in a previous round of review and you feel that this manuscript is now acceptable for publication, you may indicate that here to bypass the “Comments to the Author” section, enter your conflict of interest statement in the “Confidential to Editor” section, and submit your "Accept" recommendation.

Reviewer #1: All comments have been addressed

2. Is the manuscript technically sound, and do the data support the conclusions?

Reviewer #1: Yes

3. Has the statistical analysis been performed appropriately and rigorously? 

Reviewer #1: Yes

4. Have the authors made all data underlying the findings in their manuscript fully available?

Reviewer #1: Yes

5. Is the manuscript presented in an intelligible fashion and written in standard English?

Reviewer #1: Yes

6. Review Comments to the Author

Reviewer #1: I carefully read the authors' responses. I welcome the additional calculations and the additional data and clarifying comments: in case of FWAs mentioned by less than 10 participants and statistical analyses for all the remaining 14 PANAS labels. The suggestion for qualitative study was a suggestion for further studies.

Thanks for the corrections, I accept all of them.

7. PLOS authors have the option to publish the peer review history of their article (what does this mean?). If published, this will include your full peer review and any attached files.

Reviewer #1: **Yes: **József Rácz

---

## [Editor Report · Acceptance letter]

15 Jun 2023

PONE-D-23-04935R1 

Investigating mental representations of psychoactive substance use and other potentially addictive behaviors using a data driven network-based clustering method 

Dear Dr. File:

I'm pleased to inform you that your manuscript has been deemed suitable for publication in PLOS ONE. Congratulations! Your manuscript is now with our production department. 

Kind regards, 

on behalf of

Dr. Nicholas Aderinto Oluwaseyi 

Academic Editor

PLOS ONE